# Professionals' perspectives on interventions to reduce problematic alcohol use in older adults: a realist evaluation of working elements

Fieke A E van den Bulck  ,[1] Rikste Knijff,[1] Rik Crutzen,[2] Dike van de Mheen,[1] Rob H L M Bovens,[1,3] Sarah E Stutterheim,[2] Ien Van de Goor,[1] Andrea D Rozema[1]

**To cite:** van den Bulck FAE, Knijff R, Crutzen R, *et al*. Professionals' perspectives on interventions to reduce problematic alcohol use in older adults: a realist evaluation of working elements. *BMJ Open* 2024;**14**:e077851. doi:10.1136/bmjopen-2023-077851

¹Tranzo, Scientific Center for Care and Wellbeing, Tilburg University, Tilburg, The Netherlands
²Department of Health Promotion, Care and Public Health Research Institute (CAPHRI), Maastricht University, Maastricht, The Netherlands
³Positive Lifestyle Foundation, Nijmegen, The Netherlands

**Correspondence to**
Fieke A E van den Bulck;
f.a.e.vdnbulck@tilburguniversity.edu

## ABSTRACT

**Objectives** This study set out to understand how (which elements), in what context and why (which mechanisms) interventions are successful in reducing (problematic) alcohol use among older adults, from the perspective of professionals providing these interventions.

**Design** Guided by a realist evaluation approach, an existing initial programme theory (IPT) on working elements in alcohol interventions was evaluated by conducting semistructured interviews with professionals.

**Setting and participants** These professionals (N=20) provide interventions across several contexts: with or without practitioner involvement; in-person or not and in an individual or group setting. Data were coded and links between contexts, elements, mechanisms and outcomes were sought for to confirm, refute or refine the IPT.

**Results** From the perspective of professionals, there are several general working elements in interventions for older adults: (1) pointing out risks and consequences of drinking behaviour; (2) paying attention to abstinence; (3) promoting contact with peers; (4) providing personalised content and (5) providing support. We also found context-specific working elements: (1) providing personalised conversations and motivational interviewing with practitioners; (2) ensuring safety, trust and a sense of connection and a location nearby home or a location that people are familiar with in person and (3) sharing experiences and tips in group interventions. Furthermore, the mechanisms awareness and accessible and low threshold participation were important contributors to positive intervention outcomes.

**Conclusion** In addition to the IPT, our findings emphasise the need for social contact and support, personalised content, and strong relationships (both between client and practitioner, and client and peers) in interventions for older adults.

## INTRODUCTION

In recent decades, alcohol use has increased, particularly in older adults.[1–5] This has been accompanied by increased problematic alcohol use including high-risk drinking, binge drinking and alcohol use disorders.[6–8] Moreover, this change in drinking patterns has been associated with increased

---

**STRENGTHS AND LIMITATIONS OF THIS STUDY**

⇒ Using a realist evaluation provides in-depth insights into interventions because it can explain why interventions are successful or not.
⇒ Data were collected through interviews with a variety of professionals across 11 different alcohol interventions.
⇒ Habitation bias and acquiescence bias could have occurred in data collection, which may influence the confirmed initial programme theory.

---

alcohol-related health problems in older adults due to biological changes in metabolism, body fat, body water and the reduced ability of the liver to process alcohol,[9,10] and due to the combination of alcohol with (increasing amounts of) medication when ageing.[11]

This increase in problematic alcohol use among older adults emphasises the need for effective interventions. There are several existing approaches and specific interventions to reduce alcohol use, such as screening, brief intervention and referral to treatment, cognitive–behavioural therapy (CBT) or motivational enhancement therapy, which are considered to be appropriate for older adults.[12–16] However, most interventions to reduce (problematic) alcohol use in which older adults participate are designed for the general adult population; there is no consensus yet on what works best for specifically older adults.[17,18] Nonetheless, two systematic reviews found that effective intervention components to reduce alcohol use among older adults are personalised feedback, advice on (alcohol) behaviours, educational materials and counselling.[19,20]

Realist evaluation (RE) work from the assumption that interventions and their elements (E) work differently in different

contexts (C). The context refers to the circumstances in which interventions operate—for example, an online context in which an intervention is provided. Interventions may be successful in some contexts and not in others because the underlying processes through which the interventions bring change, called the mechanisms (M), are triggered to a different extent and lead to different intervention outcomes (O). Therefore, the interaction between the intervention elements (E), contexts (C) and mechanisms (M) plays an important role in shaping the intervention outcomes (O). The strength of RE is that it offers a means to unpack the relationships between these contexts, elements, mechanisms and outcomes. This is called the context-element-mechanism-outcome (CEMO) configuration.[21] RE starts with formulating an initial programme theory (IPT) based on the CEMOs founded in literature. The IPT then needs to be confirmed, refuted or refined using data that seeks to explain how the intervention works in real-life contexts from the perspectives of those involved in its operation. Finally, based on the new data, the programme theory might be refined.

A systematic review which included 61 studies—using RE—focused on understanding how, in what context, and why interventions are successful in preventing or reducing (problematic) alcohol consumption among older adults.[22] This review explored what works for individuals aged 18 years and older (which includes older adults), and for older adults specifically, and created an IPT based on the working elements found across six different contexts.

First, in interventions that were delivered in a context with practitioner involvement, in person and to one individual (instead of a group), the working elements of paying attention to drinking behaviour and the relationship between the client and practitioner were identified. Second, in interventions that were delivered in a context with practitioner involvement, not in person and individual, personal contact and feedback and online communication and feedback were identified. In interventions that were delivered in a context of practitioner involvement, in person and with involvement of relatives, status of the relationship and teaching the partner to deal with drinking behaviour were identified. In interventions that were delivered in a context of practitioner involvement, in-person and in-group settings, motivating to change lifestyle (delivered in a workplace setting) was identified. In interventions that were delivered in an individual context without practitioner involvement, web-based and telephone-based interventions were working elements. Finally, in interventions that were delivered in a context without a practitioner, not in person and in a group setting, focus on abstinence was identified.

Overall, the results of the review showed three general working elements: (1) providing information about the consequences of alcohol consumption; (2) personalised feedback about drinking behaviour and (3) being in contact with others and communicating with them about (alcohol) problems.[22] Of the 61 studies, only 3 evaluated interventions for older adults, and only the first and second general working elements were found in these studies.[23–25]

Evidently, in light of the small number of studies conducted specifically among older adults, more research on what works for older people is needed. The current study builds on the previous review performed using RE.

This study sets out to confirm, refute or refine the IPT, consisting of the working elements, mechanisms and outcomes across six different contexts, as identified by Boumans *et al*.[22] More specifically, the objective of this study is to understand from the perspective of professionals providing these interventions how (which elements), in what context and why (which mechanisms) interventions are successful in reducing (problematic) alcohol use among older adults. The perspective of professionals was considered important as they have extensive experience in addressing problematic alcohol use among older adults and a comprehensive understanding of what works and why. Based on the perspectives of the professionals, CEMOs of the IPT will be confirmed, refuted or refined.

## METHOD

### Study context and design

In this qualitative study, we conducted semistructured interviews with professionals providing interventions to older adults. This study is part of a larger qualitative research project investigating the perspectives of clients and close relatives of clients. Here, we focus on the professionals' perspectives. The consolidated criteria for reporting qualitative research (COREQ) guidelines were followed in reporting and design (see online supplemental material 1).[26]

### Sampling and recruitment

In the current study, we set out to recruit and interview professionals providing various interventions across different organisations throughout the Netherlands.

First, we inventoried which interventions were used in practice by searching the Dutch National Database Centre for Healthy Living (Centrum Gezond Leven) and consulting the national working groups Elderly in the Dutch Partnership for Early Detection of Alcohol Problems (Samenwerkingsverband Vroegsignalering Alcoholproblematiek) and Alcohol and Elderly of The Dutch Addiction Association (Verslavingskunde Nederland). These two working groups comprise professionals from various organisations providing different interventions.

Second, based on this inventory, we included interventions if they (1) focused on universal, selective or indicated prevention; (2) were provided in an individual and/or group setting with or without relative involvement; (3) were provided face to face, online and/or via telephone and (4) were provided in the Netherlands. Interventions were excluded if they (1) were not primarily aimed at lifestyle change; (2) were provided in a clinical setting;

(3) were group-specific interventions (eg, for veterans or athletes) or (4) focused on general health and lifestyle improvements without explicit mention of alcohol use.

Based on these selection criteria, the following Dutch alcohol interventions were included: Alcohol Information Line (Alcohol Infolijn), CBT for Substance Use (CGT bij middelengebruik), Enjoy Healthy (Gezond Genieten), Fresh Onwards, (Fris Verder), Maxx, Mirro, Moti-55, NoThanks (IkPas), Recovery Guided Personal Programme (Herstel Ondersteunend Persoonlijk Programma), Vitality Days (Vitaliteitsdagen). Snowball sampling among professionals helped us identify one other intervention: DrinkLess (MinderDrinken).

Third, we approached professionals in the two working groups working with these interventions or we approached professionals at organisations working with the interventions and asked them to connect us with professionals who were involved in providing the interventions. A convenience sampling technique was used to recruit the professionals. Interested professionals received information about the study via email and some also received information by telephone. They all received an information letter and an informed consent form. Subsequently, they were invited for an interview.

In total, we contacted 25 professionals, of whom 20 participated. The remaining five did not respond to our emails. Participants' characteristics are shown in table 1. Characteristics of included interventions are shown in table 2.

### Data collection
Based on the participants preferences, online (80%) and telephone (20%) interviews were conducted between June and December 2022, each lasting an average of 59.3 min (ranging from 40.4 to 85.4 min). Demographic variables including age, gender and some questions about the intervention were asked prior to the interviews. All interviews were preceded by obtaining oral informed consent. We used an interview guide that measured the perceived outcomes of the interventions, the working elements and mechanisms related to the outcome(s), and the influence of the setting in which the interventions were provided. At the end of the interview guide, we presented the previous found CEMO configurations of the IPT of the review of Boumans et al[22]; we invited participants to confirm, refute or refine the CEMOs and asked if they could comment on the CEMOs (see interview guide, online supplemental material 2). Interviews were conducted until data saturation occurred.

### Patient and public involvement
None.

### Data analysis
Professional transcriptionists transcribed the interviews verbatim. Based on the IPT, FAEvdB drafted a code tree including CEMO configurations. Researchers FAEvdB and RK independently and, in parallel, coded 6 of the 20

interviews based on the RE approach, which means that links between contexts, elements, mechanisms, outcomes were sought within the data. After the first transcript was coded by FAEvdB and RK, FAEvdB cross-checked both their configurations and codes. FAEvdB and RK iteratively

**Table 1** Participants' characteristics

| Baseline characteristic | Participants n=20 |
|---|---|
| Gender | |
| Woman | 15 |
| Man | 5 |
| Age (years) | |
| 25–39 | 6 |
| 40–54 | 4 |
| 55–69 | 7 |
| ≥70 | 3 |
| Experience with the intervention (in years) | |
| <1 | 2* |
| 1 | 3 |
| 2 | 4 |
| 3 | 2 |
| 4 | 4 |
| 5 | 3 |
| 6 | 1 |
| 7 | – |
| 8 | – |
| 9 | 1 |
| Role† | |
| Healthcare worker, prevention worker, coach, psychiatric mental health nurse or psychologist | 13 |
| Peer support worker | 1 |
| Volunteer | 3 |
| Founder of intervention | 3 |
| Advisor, coordinator, manager | 3 |
| Intervention | |
| Alcohol Information Line | 3 |
| CBT for Substance Use | 2 |
| DrinkLess | 1 |
| Enjoy Healthy | 1 |
| Fresh Onward | 4 |
| Maxx | 1 |
| Mirro | 1 |
| Moti-55 | 2 |
| NoThanks | 2 |
| Recovery Guided Personal Programme | 1 |
| Vitality Days | 2 |

*One professional was only involved in the development of the intervention.
†Some professionals had multiple roles in the same interventions—for example, working as a coach and coordinator.
CBT, cognitive–behavioural therapy.

**Table 2** Intervention characteristics

| Intervention | Contact in intervention with | | | Setting | | | Target group intervention | |
| | Professional | Peers | Relatives | Online, telephone | Face to face | Sessions | Client's alcohol use* | Older adults (55+ or 65+) |
| --- | --- | --- | --- | --- | --- | --- | --- | --- |
| Alcohol Information Line | X | | | X | | ≥1 | I | |
| CBT for Substance Use | X | X | X† | | X | >1 | IV | X |
| DrinkLess | X† | | | X | | N/a | III | |
| Enjoy Healthy | X | X | | | X | 1 | I | X |
| Fresh Onwards | X | X | X† | | X | >1 | IV | X |
| Maxx | | | | X | | N/a | III | |
| Mirro | | | | X | | N/a | III | |
| Moti-55 | X | | X† | | X | >1 | II | X |
| NoThanks | X† | | | X | | N/a | III | |
| Recovery Guided Personal Programme | X | | X† | | X | >1 | V | |
| Vitality Days | X | X† | X† | | X | 1 | I | X |

*I=People with all types of alcohol use patterns; II=people with early-stage problematic alcohol use; III=people with all types of alcohol use patterns, except for (heavy) problematic alcohol use or alcohol addiction; IV=people with problematic alcohol use and alcohol addiction; V=people who are in recovery from addiction.
†The involvement of this person in the intervention is optional and according to the client's preference.
CBT, cognitive–behavioural therapy; N/a, not available.

discussed differences in configurations or coding, and, where relevant, refinements were made. These steps were repeated for all of the six initial interviews, at which time coding was fully aligned. Thereafter, FAEvdB coded the remaining interviews independently. When in doubt about how best to code sections of the data, FAEvdB consulted with RK and ADR. Additionally, to improve the quality of the coding further, RK, RC, RHLMB, IVdG, SES and ADR reviewed the formed codes and CEMOs iteratively and provided feedback. Codes were then adjusted if necessary and some new codes were added. Data were analysed by using Atlas.TI.

## RESULTS
The results are categorised according to the context in which the interventions are delivered: (1) involvement or lack of involvement of a practitioner who provides the intervention directly to the participant; (2) provision of the intervention in person or not and (3) individual treatment, group treatment or treatment with relatives' involvement. We found data across six combinations of contexts to which the IPT conformed. Below, we describe the working intervention elements (E). Table 3 provides an overview of the key working elements, mechanisms and outcomes, mentioned by more than one participant. Table 4 provides an overview of the summarised findings of the programme theory. For an even more comprehensive table including all CEMOs, even those mentioned by just one respondent (see online supplemental table 1).

### Practitioner—in-person—individual
Professionals confirmed that (1) paying attention to drinking behaviour (E) is important in interventions to reduce alcohol use (O) because it could stimulate clients to start thinking about their alcohol use (M) and create awareness of their use (M).

> Alcohol is, of course something that has a risk, and can cause health damage. People think about that. 'Do I want to do that to myself? I'm getting older, I'm getting more forgetful.' So they start to think a lot more about what they are doing to themselves because of alcohol use. If you actually tell them that as a professional, then people really think twice about it: 'Wait a minute, I still want to enjoy my life and I want to age healthily'. [R9, prevention worker]

They also emphasised the importance of (2) paying attention to lifestyle (E) and (3) using specific communication approaches (E), such as motivational interviewing. The professionals confirmed that (4) the relationship between client and practitioner (E) is important because it could trigger open communication (M), and collaboration between client and practitioner that improves the relationship between practitioner and client (M). Finally, they mentioned that a (5) favourable setting (E) in which the intervention is provided is important—that is, a location that is nearby because it contributes to low threshold participation (M).

### Practitioner—not in-person—individual
Professionals confirmed that personal contact combined with feedback reduced alcohol use (O). In addition, they mentioned several other (1) communication approaches (E), such as motivational interviewing, personalised contact and anonymous participation. These approaches triggered different mechanisms—that is, tailoring (M), and low threshold and flexible participation (M).

**Table 3**  The programme theory

| Context (C) | Element of intervention (E) | Mechanism (M) | Outcome (O) | Target group* | IPT† |
|---|---|---|---|---|---|
| A. Practitioner—in-person—individual | 1. Paying attention to drinking behaviour | | | | |
| | 1.1. Paying attention to drinking behaviour‡ | Think and act differently about alcohol use | Less or no alcohol use | X | ✓ |
| | 1.2. Tracking alcohol use and reflecting on use | a. Creating insight into use | Less or no alcohol use | II, III, IV, X | + |
| | | b. Thinking about alcohol use | Awareness | | |
| | 1.3. Paying attention to abstinence, coping planning and practising with abstinence | Raising awareness of alcohol use | | I, IV, X | + |
| | 1.4. Pointing out the risks and consequences of drinking behaviour | Thinking about indirect consequences of use | Awareness, less or no alcohol use | I, II, IV | + |
| | 2. Paying attention to lifestyle | | | | |
| | 2.1. Paying attention to meaning of life and teaching lifestyle changes | | Less or no alcohol use | II | + |
| | 3. Communication approaches | | | | |
| | 3.1. Motivational interviewing | | | II, X | + |
| | 3.2. Personalised content and conversation | | | I, II | + |
| | 4. The relationship between the client and practitioner | | | | |
| | 4.1. Safety, trust, connection and not being judged in contact with professional | Having open communication | Less alcohol use | II, X | + |
| | 4.2. Empathic behaviour of practitioner‡ | Collaborating in the identification of help and needs, and improving the relationship between client and practitioner | Less or no alcohol use | I, X | ✓ |
| | 4.3. Critical, controlling and confronting behaviour of practitioner | | | II, IV, X | + |
| | 5. Favourable setting | | | | |
| | 5.1. Location is nearby home and/or familiar | Low threshold participation | | I, IV | + |
| B. Practitioner—not in-person—individual | 1. Communication approaches | | | | |
| | 1.1. Contact always possible | Accessible help in difficult situations | | I, III | + |
| | 1.2. Conversation over the phone | Accessible and flexible participation | | I, X | + |
| | 1.3. Anonymous | a. Distance in contact, making client more likely to accept advice | Think differently about alcohol use | I | + |
| | | b. Low threshold | | | |
| | | c. Honest instead of socially desirable contact | | | |
| | 1.4. Motivational interviewing | | Awareness | I, X | + |
| | 1.5. Personalised content and conversation | a. Tailoring | | I, III | + |
| | | b. Non-commitment disappears | Less or no alcohol use | I, III | + |
| | | c. Emotional support | | I, III, X | + |
| | 1.6. Personal contact and feedback‡ | | Less or no alcohol use | X | ✓ |
| | 1.7. Online communication and feedback‡ | | Less or no alcohol use | | – |
| | 2. The relationship between the client and practitioner | | | | |
| | 2.1. Practitioner is empathic, supportive and listens | Engaging, open communication and client accepts advice or information from practitioner | | I | + |
| | 3. Providing additional help | | | | |
| | 3.1. Personal treatment with practitioner additional to self-help | Extra attention, tailoring | | III, X | + |

Continued

**Table 3** Continued

| Context (C) | Element of intervention (E) | Mechanism (M) | Outcome (O) | Target group* | IPT† |
|---|---|---|---|---|---|
| C. Practitioner—in-person—relatives | 1. Teaching the partner to deal with drinking behaviour | | | | |
| | 1.1. Teaching the partner to deal with drinking behaviour‡ | More understanding and support from the relative for the client | | I, IV, X | ✓ |
| | 2. Support of relatives | | | | |
| | 2.1. Relative provides support | | Less or no alcohol use | X | + |
| D. Practitioner—in-person—group component | 1. Paying attention to drinking behaviour | | | | |
| | 1.1. Pointing out the risks and consequences of drinking behaviour | | | IV, V | + |
| | 1.2. Paying attention to abstinence and practicing with abstinence | | Less or no alcohol use | IV | + |
| | 2. Paying attention to lifestyle | | | | |
| | 2.1. Motivating to change lifestyle‡ | | | | ~ |
| | 2.2. Attention and tips regarding meaning of life, lifestyle and problems when ageing | Motivation for participation and change | Less or no alcohol use, know how to improve life quality during abstinence | IV, X | + |
| | 2.3. Motivation to change lifestyle delivered in a workplace setting‡ | | | | ~ |
| | 2.4. In a workplace setting and paying attention to prevention and lifestyle | | | X | + |
| | 3. Communication approaches | | | | |
| | 3.1. Motivational interviewing | | Less or no alcohol use, awareness | IV, X | + |
| | 3.2. Personalised content and conversation | | | IV, X | + |
| | 4. The relationship between the client and peers | | | | |
| | 4.1. Contact with peers | Recognition, connection and support | | IV, V, X | + |
| | 4.2. Closed group and maximum group size | a. Create an emotionally safe space | | I, IV | + |
| | | b. More attention per client | | | |
| | 4.3. Engagement, understanding and support towards peers | Fellowship, connection and safety | Less or no alcohol use | IV | + |
| | 4.4. Addressing peers | Gain new insights | Less or no alcohol use | II, III, IV, V | + |
| | 4.5. Sharing experiences and tips with peers | a. Being hopeful in own process | | IV, V, X | + |
| | | b. Feeling recognised, relativising own situation | | | |
| | 5. The relationship between the client and practitioner | | | | |
| | 5.1. Open attitude, not being judged and accessible contact with practitioner | | | V | + |
| | 6. Other activities | | | | |
| | 6.1. Having lunch after every session | Making connection and pleasant atmosphere | | IV, V | + |
| | 7. Favourable setting | | | | |
| | 7.1. Pleasant and relaxed atmosphere | | | I, IV | + |
| | 7.2. Location is nearby home and/or familiar | Low threshold participation | | I, IV | + |
| E. No practitioner—not in-person—individual | 1. Paying attention to drinking behaviour | | | | |
| | 1.1. Tracking alcohol use | Insight into use and tracking progress | Less or no alcohol use | III, X | + |
| | 1.2. Abruptly quitting alcohol use | Positive learning experience | | II, III | + |
| | 1.3. Pointing out the risks and consequences of drinking behaviour | | | III | + |

Continued

van den Bulck FAE, *et al. BMJ Open* 2024;**14**:e077851. doi:10.1136/bmjopen-2023-077851

**Table 3** Continued

| Context (C) | Element of intervention (E) | Mechanism (M) | Outcome (O) | Target group* | IPT† |
|---|---|---|---|---|---|
| | 2. Using tools | | | | |
| | 2.1. Web-based and telephone-based interventions‡ | | | | ~ |
| | 2.2. Online self-help tool | Accessible, low threshold and controlled participation | Awareness, less or no alcohol use, going to treatment | III, X | + |
| | 2.3. Via telephone | | | X | + |
| | 2.4. Regular newsletter | Being actively engaged | | III | + |
| | 3. The relationship between the client and peers | | | | |
| | 3.1. Joining with others, contact with peers | Fellowship, recognition and support | | III | + |
| | 4. Favourable setting | | | | |
| | 4.1. Participation independent of location | Accessible and low threshold | | III | + |
| | 5. Additional help | | | | |
| | 5.1. Personalised content and help additional to self-help | | | III | + |
| F. No practitioner—not in-person—group component | 1. Paying attention to drinking behaviour | | | | |
| | 1.1. Intervention to abstinent people‡ | | | | ~ |
| | 1.2. Focus on abstinence and paying attention to withdrawal | | | X | + |
| | 2. Self-help groups | | | | |
| | 2.1. AA and self-help group | | | X | + |
| | 3. The relationship between the client and peers | | | | |
| | 3.1. Contact with peers | | | IV, X | + |
| | 3.2. Sharing experiences and tips with peers | | | IV, X | + |

*Target group of the intervention. I=People with all types of alcohol use patterns; II=people with early-stage problematic alcohol use; III=people with all types of alcohol use patterns, except for (heavy) problematic alcohol use or alcohol addiction; IV=people with problematic alcohol use and alcohol addiction; V=people who are in recovery from addiction. X=No target group mentioned because these elements were derived from CEMO configuration that we presented as statements during the interviews.
†✓=Initial theory confirmed, ~=initial theory refined, +=new theory added−=initial theory not confirmed.
‡Elements of initial program theory.
IPT, initial programme theory.

And certainly with older people because they often already have certain habits for years or perhaps have a bad marriage or a very stressful job. So those life-stage issues, you can sort those out in personal treatment. [R3, psychologist and advisor]

It also makes it low threshold for people to call. We often have people on the phone who admit for the first time at all that something is going on. [...] For example, people often call in their car and I think that is also a nice anonymous place where no one, no family member, a colleague, can see who they are calling. [...] People can choose when and where they have this conversation, so they can choose a time and location that they prefer. [R4, health care worker]

The professionals added that (2) the relationship between client and practitioner (E) is important—for example, when the professional is empathic, supportive and listens to the client—because this could trigger engagement (M), followed by open communication where the client accepts advice or information from the practitioner (M). Finally, they mentioned that (3) providing additional help (E) can

be important because it could provide extra attention (M) and tailoring (M).

**Practitioner—in-person—relatives**
Professionals confirmed that a partner is more understanding and supportive of the client (M) when (1) the partner is taught to deal with the drinking behaviour of the client (E), and that (2) support by relatives (E) can lead to alcohol reduction (O) among clients.

It is important for them to understand how addiction works, that it's not solved at once and that it is actually a disease. [...] . So, the relative regains a bit more confidence and yes, this often creates space in which they can grow together and take steps. On the one hand, this strengthens the client and, on the other hand, it gives a relative more confidence that it can be done. [R16, peer support worker]

**Practitioner—in-person—group component**
Professionals mentioned that (1) paying attention to drinking behaviour (E) is important because it contributes to a reduction in alcohol use (O). They confirmed

**Table 4** Summary of the programme theory

| Context (C) | Element of intervention (E) | Mechanism (M)* | Outcome (O) |
|---|---|---|---|
| A. Practitioner—in-person—individual | Paying attention to drinking behaviour | Changing perceptions of own use (1.1., 1.2.a., 1.2.b., 1.3.) and perceptions of consequences (1.4.) | Less or no alcohol use, awareness |
| | Paying attention to lifestyle | | |
| | Communication approaches | | |
| | The relationship between the client and practitioner | Improving communication (4.1.) and collaboration with practitioner (4.2.) | |
| | Favourable setting | Low threshold participation (5.1.) | |
| B. Practitioner—not in-person—individual | Communication approaches | Low threshold participation (1.3.b.), tailoring (1.1., 1.2., 1.5.a.), improving communication with practitioner (1.3.a., 1.3.c.), engagement (1.5.b) and emotional support (1.5.c) | Less or no alcohol use, awareness, changing perspective on own alcohol use |
| | The relationship between the client and practitioner | Engagement and improving communication with practitioner (2.1.) | |
| | Providing additional help | Tailoring (3.1.) | |
| C. Practitioner—in-person—relatives | Teaching the partner to deal with drinking behaviour | More understanding and support from the relative for the client (1.1.) | Less or no alcohol use |
| D. Practitioner—in-person—group component | Paying attention to drinking behaviour | | Less or no alcohol use, awareness |
| | Paying attention to lifestyle | Motivation (2.2.) | |
| | Communication approaches | | |
| | The relationship between the client and peers | Social and emotional support (4.1., 4.3., 4.5.b.), pleasant atmosphere (4.2.a.), tailoring (4.2.b.) and changing perceptions of own use (4.4., 4.5.a., 4.5.b.) | |
| | The relationship between the client and practitioner | | |
| | Other activities | Social contact and pleasant atmosphere (6.1.) | |
| | Favourable setting | Low threshold participation (7.2.) | |
| E. No practitioner—not in-person—individual | Paying attention to drinking behaviour | Changes perceptions on own use (1.1., 1.2.) | Less or no alcohol use |
| | Using tools | Low threshold (2.2.) and engagement (2.4.) | |
| | The relationship between the client and peers | Social and emotional support (3.1.) | |
| | Favourable setting | Low threshold participation (4.1.) | |
| | Additional help | | |
| F. No practitioner—not in-person—group component | Paying attention to drinking behaviour | | |
| | Self-help groups | | |
| | The relationship between the client and peers | | |

*The mechanisms that are presented are the summarised mechanisms from table 3, along with the numbering of the original mechanisms as indicated in table 3.

that (2) paying attention to lifestyle (E) was also a working element in this context because it motivates people to participate in the intervention and to change (M), which ultimately contributes to reduced alcohol use (O). In addition, they mentioned that (3) communication approaches (E), including motivational interviewing, and personalised contact and conversations were important. Professionals also mentioned that (4) the relationship between client and peers (E) plays an important role in interventions because it could provide recognition,

connection, support, hope, safety and new insights (M) for the client.

> There are often a lot of negativities in the family atmosphere and I think in a group with peers it is accepted that people have a problem and they start looking to get it under control. So, there is engagement, an open and accepting attitude that makes people feel safe, and I think that is very important in a group, that people feel very safe to talk about it without feeling judged. [R13, MH psychologist]

Again, (5) the relationship between the client and practitioner (E)—for example, when the professional has an open attitude and does not judge the client—was important in interventions according to the professionals. Providing (6) other activities (E) such as eating after a joint session was helpful because these contribute to making a connection (M) and a pleasant atmosphere (M).

Finally, (7) a favourable setting was important because it contributed to low threshold participation.

> It is often a very high threshold for people to go to an addiction center. So, I think that this kind of treatment could also be given in a community center, or a nursing home, or at the GP in a small group, or something. [R14, psychiatric mental health nurse]

### No practitioner—not in-person—individual

Professionals mentioned that (1) paying attention to drinking behaviour (E) contributes to reductions in alcohol use (O). Also (2) using tools (E), such as an online self-help tool, was considered beneficial in this setting because it enables low threshold participation (M) for participants and allowed them to be actively involved (M). Again, (3) the relationship between client and peers (E) was important because it provided clients with fellowship, recognition and support (M).

> I think that for this target group it is very welcome that, additionally in an online setting, they can do it alone or with a friend, or with a group where you meet once per week or every two weeks. [RP6, founder of intervention and manager]

Also, (4) a favourable setting (E) was important because it could make participation low threshold (M). Finally, (5) additional help (E)—that is, providing personalised content and help additional to self-help—was also seen as effective.

### No practitioner—not in-person—group component

Professionals mentioned that (1) paying attention to drinking behaviour (E), (2) self-help groups (E) and (3) the relationship between the client and peers (E) were all important in interventions in this context. No mechanisms were found that were mentioned by more than one professional.

## DISCUSSION
### Key findings

Overall, five elements that were mentioned by more than one respondent were found in at least three contexts and can be labelled as general working elements: (1) pointing out the risks and consequences of drinking behaviour; (2) paying attention to abstinence; (3) promoting contact with peers; (4) providing personalised content and (5) providing support. We also found context-specific working elements, namely providing personalised conversations and motivational interviewing, specifically in interventions with practitioner involvement; ensuring safety, trust and a sense of connection and a location nearby home or a location that people are familiar with in interventions in person; and sharing experiences and tips in interventions in group settings. Moreover, two mechanisms that were mentioned by more than one respondent were found in at least three contexts: awareness and accessible and low threshold participation. Our findings are generally in line with the IPT. However, we found many additional working elements, mechanisms and outcomes that were posited by several professionals, resulting in a refined programme theory.

### Interpretation of findings

The first addition to the IPT is that older adults need social contact and support. Promoting contact with peers can help reduce loneliness or social isolation, and providing support by a practitioner, peers or relatives can help deal with negative emotions and problematic alcohol use. Compared with the general adult population, older adults are more often facing retirement, illness, loss of relatives, loneliness or hopelessness, which may facilitate negative health consequences of alcohol use.[27 28] It is important to facilitate the process of helping older adults to develop contacts—for example, in leisure settings in a location that is nearby home and/or familiar—to make participation low threshold.

The second addition to our IPT is that not only personalised feedback, but also providing personalised content and conversations is helpful for older adults, which is in line with findings in previous research showing that a personal approach with personalised feedback[23–25] and personalised reports[23 24] is effective for older adults. Understanding the client's motives for alcohol use can help providers to be more specific and effective in providing personalised and tailored interventions. A tailored approach with consideration of the client's needs, with aging-related challenges taken into account, appears to be very effective.[29]

The third addition to our IPT is that both the relationship between the client and the practitioner and the relationship between the client and peers are important in interventions. Previous research has shown that therapeutic alliance in the relationship between the client and the practitioner is associated with a reduced likelihood of dropout.[30] Furthermore, older adults with substance use problems prefer interventions that are led by warm,

honest, caring and relatively non-confrontational providers.[31] These findings are in line with our working element ensuring safety, trust, and a sense of connection and the importance of the relationship between client and provider.

In contrast, previous research has shown mixed findings on the relationship between the client and peers in substance use disorder populations: some research found a significant association between group cohesion and intervention outcomes, and group processes and outcomes,[32–37] while other research found no significant association between group cohesion and outcomes.[38 39] Even though there seems to be no consensus in the literature on the association between the relationship between client and peers and intervention outcomes for the general adult population, we argue that there could be a strong association for older adults. Age-specific treatments may work better than mixed-age treatments with older adults.[40] A group composition that matches the preferences of older adults could be a prerequisite to ensure safety, trust and a sense of connection, contributing to positive outcomes.

Finally, we found two important mechanisms that contribute to positive intervention outcomes. First, awareness is an important mechanism in preventing or reducing problematic alcohol use. A previous study by Heijkants et al[41] reported that older adults have limited awareness of the negative health effects of alcohol, and that the majority of older adults in their study did not consider alcohol to be addictive or unhealthy. In addition, only a small proportion of older adults know the current standard for lower risk alcohol consumption,[41] and people with problematic alcohol use often do not recognise their own drinking pattern as problematic.[42 43] This lack of knowledge emphasises the importance among older adults of the working element pointing out the risks and consequences of drinking behaviour. Emphasising these risks can be experienced as threatening, and therefore, it is important to enhance older adults' self-efficacy in the intervention first.[44] Furthermore, knowledge is a fundamental first step, but information provision alone is not sufficient for behaviour change.[45] Second, accessible and low threshold participation form an important mechanism in interventions for older adults. A major barrier for older adults seeking help is that they do not want to be labelled an 'alcoholic' or 'addict',[41] which can make participation in interventions that are provided in addiction care centres high threshold. Providing interventions in a location that people do not directly associate with alcoholism—for example, a community centre—can make participation low threshold. Another barrier for older adults can be difficulties with transportation.[31] Travel allowances and accessible interventions can help older adults overcome these barriers.

## Limitations and strengths

This study has some limitations. In the Netherlands, people with lower socioeconomic status and ethnic minorities are often not reached by lifestyle intervention efforts and show low participation rates.[46 47] It is possible that the interventions included in this study are less likely to reach these groups, which could mean that the professionals have limited experiences with these subgroups and that the overview of working elements and mechanisms is not fully adequate for these subgroups. It is, therefore, important to provide interventions with personalised content and conversations for specific subgroups, because this can provide a tailored approach. For example, older adults with a language deficiency could benefit from more specific elements in communication with the professional. Also, cultural differences may ask for different approaches. A study showed that religious older Moroccan-Dutch and Turkish-Dutch women can have other motivators (eg, religious or cultural) to abstain from alcohol use than older native Dutch women.[48] Further research should focus on what works for specific subgroups of older adults and to what extent these older adults participate in interventions to reduce problematic alcohol use. The second limitation concerns biases when the CEMO statements were presented at the end of the interview. It is possible that acquiescence bias occurred here, meaning that respondents perhaps agreed more easily to these statements because they responded in ways they thought the researcher wanted them to respond, or because they wanted to finish the interview. Respondents were invited to explain their opinions on these statements, but they often responded only by saying that they agreed with these statements without explanation. Habituation bias could also have occurred when the participants provided the same answers in response to similarly worded CEMO statements. These two biases may influence our confirmed IPT. Finally, some CEMO configurations were only confirmed when the statements were presented. Some working elements—for example, empathic behaviour of the practitioner—might be so obvious that they were not initially mentioned. We think that there could be more such elements that are missing in our overview because of these factors.

Strengths of this study are that we were able to include a variety of professionals across all 11 interventions and the high response rate. Another strength is the RE approach used, as this provides explanations from the perspective of professionals of what works to reduce problematic alcohol use among older adults. This approach provides in-depth insights into interventions because it can explain why interventions are successful or not.

## Practical and research implications

Based on our findings, we have a number of recommendations for practice and future research. Professionals and developers of interventions might focus on three aspects. First, practitioners should pay attention to the elements related to the role of the practitioner in in-person interventions, more specifically in existing brief interventions which are already often used by practitioners. Second, developers of interventions need to

pay particular attention to accessible or low threshold participation—for example by providing interventions in a favourable setting. Third, practitioners and developers together should focus more on the element promoting contact with peers. For example, practitioners working in group interventions can invite clients to share experiences and tips with peers, and developers of online self-help tools can refer people to online forums where people can share experiences and tips with peers.

Further research should focus on the perspectives of clients who participate in interventions and their relatives to refine the programme theory. Also, further research should focus on interventions in the context with a practitioner, not in-person and individual and in the context with no practitioner, not in-person and in groups, because elements in these contexts are limited in this study.

## CONCLUSION

In addition to the previous formulated IPT, our findings emphasise the need for social contact and support, personalised content and strong relationships (between client and practitioner, and client and peers) in interventions for older adults. Whereas the working element 'social contact' was not found as a working element in previous literature, probably due to the limited number of studies performed, our study found this to be an additional important working element, and providers of alcohol interventions should be aware of and make use of this element when treating older adults. Future research should study the perspective of older adults themselves and their relatives on working elements and mechanisms in interventions.

**Contributors** FAEvdB, ADR, RC, RHLMB and DvdM developed the research questions and study methodology. FAEvdB and ADR developed an interview guide an RC, RHLMB and DvdM provided inputs to the interview guide. Data were collected by FAEvdB. Data were coded by FAEvdB and co-coded by RK and checked by ADR. ADR, RC, RHLMB, DvdM, IVdG and SES then provided feedback on the code document. FAEvdB produced the draft manuscript under daily supervision of ADR. ADR, RC, RHLMB, DvdM, IVdG and SES have revised the manuscript critically for important content. ADR acted as the guarantor for the overall content.

**Funding** This research was funded by The Netherlands Organisation for Health Research and Development (ZonMw number: 555002034).

**Competing interests** None declared.

**Patient and public involvement** Patients and/or the public were not involved in the design, or conduct, or reporting, or dissemination plans of this research.

**Patient consent for publication** Not applicable.

**Ethics approval** This study involves human participants and this study was approved by the Ethics Review Board at Tilburg University (RP508). Participants gave informed consent to participate in the study before taking part.

**Provenance and peer review** Not commissioned; externally peer reviewed.

**Data availability statement** No data are available.

**ORCID iD**
Fieke A E van den Bulck http://orcid.org/0009-0002-2247-8446

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
