## [Reviewer comments · BMJ Open]

ARTICLE DETAILS

TITLE (PROVISIONAL)	Professionals' perspectives on interventions to reduce problematic alcohol use in older adults: A realist evaluation of working elements.
AUTHORS	van den Bulck, Fieke; Knijff, Rikste; Crutzen, Rik; van de Mheen, D; Bovens, Rob; Stutterheim, Sarah; Van de Goor, Ien; Rozema, A

VERSION 1 – REVIEW

REVIEWER	Morris, James London South Bank University
REVIEW RETURNED	22-Sep-2023

GENERAL COMMENTS	Thank you for the opportunity to review 'Professionals' perspectives on interventions to reduce and prevent problematic alcohol use in older adults: A realist evaluation of working elements.' I found this to be an interesting study addressing an important issue of alcohol use amongst older adults. My main comment is that overall I found the manuscript somewhat hard to follow, even after several readings. In particular, the narrative organization/categorization of the interventions took me some time to comprehend (the tables were easier to follow). As such I think the most significant improvement to this manuscript would be to try and reconfigure the presentation of the 'six combinations of contexts' such that these are easier to comprehend. Whilst the manuscript makes some references to motivational interviewing, many of the interventions and related elements seem very relevant to an extensive 'brief interventions' literature (albeit existing in many forms/ names e.g, predominantly 'Screening and Brief Interventions and Referral to Treatment' in the US). Some further reference to the relevance of these findings to the brief intervention literature I think should be incorporated within the introduction and implications. Under 'Interpretation of findings', the authors state there are two important mechanisms that contribute to positive outcomes. The first is identified as 'awareness' but is not clear what the second is (perhaps knowledge?). Concerning the mechanism of awareness, some recent conceptual and empirical work has focused on this (termed 'problem recognition') in non-help seeking drinking groups [1, 2] which it may be worth highlighting. Notably, one recent study on problem recognition [3] also empirically supports the important point made about the threat of 'alcoholic' labeling undermining 'awareness'/problem recognition.
--

	[1] James Morris, Ian P. Albery, Antony C. Moss, Nick Heather, Chapter 10 - Promoting problem recognition amongst harmful drinkers: A conceptual model for problem framing factors, Editor(s): Daniel Frings, Ian P. Albery, The Handbook of Alcohol Use, Academic Press, 2021 https://doi.org/10.1016/B978-0-12-816720-5.00026-8 [2] Jessica J. Smith, Panagiotis Spanakis, Rachael Gribble, Sharon A.M. Stevelink, Roberto J. Rona, Nicola T. Fear, Laura Goodwin, Prevalence of at-risk drinking recognition: A systematic review and meta-analysis, Drug and Alcohol Dependence, Volume 235, 2022, https://www.sciencedirect.com/science/article/pii/S0376871622001867 [3] J. Morris, A.C. Moss, I.P. Albery, N. Heather, The "alcoholic other": Harmful drinkers resist problem recognition to manage identity threat, Addictive Behaviors, Volume 124, 2022, https://doi.org/10.1016/j.addbeh.2021.107093.
--	---

REVIEWER	Fyffe, Ian Fraser Health Authority, Long Term Care and Assisted Living Research Unit
REVIEW RETURNED	03-Dec-2023

GENERAL COMMENTS	First off I wanted to state that this is important work in an underdeveloped area. I had a few concerns about the paper though:  1) Please make the research question more explicit so that readers can find it quickly and easily. 2) Please describe the Initial Program Theory of Boumans et al in greater detail that you refer to. Readers will likely be unfamiliar with the work and it is problematic not describing it in greater detail on line 55. 3) Please fix the formatting in tables 2 and 3 so that they have greater readability. Table 2 had "professional" across three lines, this was the same for "alcohol use" and "relatives". This also goes for Table 3 such as "target group intervention". 4) In the results section, please label each italicized component as either a context, element, mechanism, or outcome. I believe that most of these are elements, but it would assist the reader in better understanding what each component is without confusing elements for mechanisms for example. This could be easily changed by writing for example on Line 51 "Professionals confirmed that the element (1) paying attention to drinking behavior..." 5) Please define context, element, mechanism and outcome better on lines 44-49 for readers unfamiliar with this methodology. You put them together nicely but they still lack robust definitions. 6) Table 3 is great, I just wanted to commend you all on how you put that information together.
--

VERSION 1 – AUTHOR RESPONSE

Reviewer #1

Thank you for the opportunity to review 'Professionals' perspectives on interventions to reduce and prevent problematic alcohol use in older adults: A realist evaluation of working elements.' I found this to be an interesting study addressing an important issue of alcohol use amongst older adults.

Comment 4: My main comment is that overall I found the manuscript somewhat hard to follow, even after several readings. In particular, the narrative organization/categorization of the interventions took me some time to comprehend (the tables were easier to follow). As such I think the most significant improvement to this manuscript would be to try and reconfigure the presentation of the 'six combinations of contexts' such that these are easier to comprehend.

Reply: We thank the reviewer for the comments and suggestion. We try to present the contexts more easier. First, we made a summarized table of our program theory to provide a better overview of our findings to ensure that the categorization of contexts is easier to comprehend. Second, we added some sentences in the introduction with extra explanations and clarifications of the contexts. Third, we mentioned in the result section that the contexts we describe are the same contexts that the IPT conformed to.

Revised text:

"Table 4 provides an overview of the summarized findings of the program theory." (See page 14 and 15 for Table 4).

Table 4. Summary of the program theory

Context	Element of intervention	Mechanism*	Outcome
A. Practitioner – in-person – individual	Paying attention to drinking behavior	Changing perceptions on own use (1.1, 1.2.a., 1.2.b., 1.3.) and perceptions consequences (1.4.)	Less or no alcohol use, awareness
	Paying attention to lifestyle Communication approaches The relationship between the client and practitioner Favorable setting	Improving communication (4.1.) and collaboration with practitioner (4.2.) Low-threshold participation (5.1.)	
	B. Practitioner – not in person – individual	Communication approaches	Low- threshold participation (1.3.b), tailoring (1.1., 1.2., 1.5.a.), improving communication with practitioner (1.3.a., 1.3.c.), engagement (1.5.b), and emotional support (1.5.c)
	The relationship between the client and practitioner Providing additional help	Engagement and improving communication with practitioner (2.1.) Tailoring (3.1.)	
C. Practitioner —in- person— relatives	Teaching the partner to deal with drinking behavior	More understanding and support from the relative for the client (1.1.)	Less or no alcohol use
D. Practitioner —in- person— group component	Paying attention to drinking behavior		Less or no alcohol use, awareness
	Paying attention to lifestyle Communication approaches The relationship between the client and peers	Motivation (2.2.) Social and emotional support (4.1., 4.3., 4.5.b.), pleasant atmosphere (4.2.a.), tailoring (4.2.b.), and changing perceptions on own use (4.4., 4.5.a., 4.5.b.)	
	The relationship between the client and practitioner Other activities	Social contact and pleasant atmosphere (6.1.)	

	Favorable setting	Low-threshold participation (7.2.)	
E. No practitioner—not in-person—individual	Paying attention to drinking behavior Using tools The relationship between the client and peers Favorable setting Additional help	Changes perceptions on own use (1.1., 1.2.) Low-threshold (2.2.) and engagement (2.4.) Social and emotional support (3.1.) Low-threshold participation(4.1.)	Less or no alcohol use
F. No practitioner – not in-person – group – component	Paying attention to drinking behavior Self-help groups The relationship between the client and peers		

*= The mechanisms that are presented are the summarized mechanisms from Table 3, along with the numbering of the original mechanisms as indicated in Table 3

“First, in interventions that were delivered *in the context* [...] Second, in interventions that were delivered *in the context* [...] Third, in interventions that were delivered *in the context* [...] Fourth, in interventions that were delivered *in the context* [...] Fifth, in interventions that were delivered *in the context* [...] Sixth, in interventions that were delivered *in the context*” (See page 3 and 4)

“This study is set out to confirm, refuse, or refine the IPT, *consisting of the working elements, mechanisms and outcomes across six different contexts, as identified by Boumans et al. (22). More specifically, the objective of this study is to* understand from the perspective of professionals providing these interventions how (i.e., which elements), in which context, and why (which mechanisms) interventions are successful in reducing (problematic) alcohol use among older adults.” (See page 4)
“We found data across six combinations of contexts that the IPT conformed to.” (See page 9)

Comment 5: Whilst the manuscript makes some references to motivational interviewing, many of the interventions and related elements seem very relevant to an extensive 'brief interventions' literature (albeit existing in many forms/ names e.g, predominantly 'Screening and Brief Interventions and Referral to Treatment' in the US). Some further reference to the relevance of these findings to the brief intervention literature I think should be incorporated within the introduction and implications.

Reply: We thank the reviewer for the suggestion. We acknowledge the relevance of SBIRT for older adults in reducing problematic alcohol use. We agree with the suggestion that we should include the relevance of SBIRT in our introduction, and therefore we mentioned SBIRT as one of the existing approaches that are appropriate for older adults. Additionally, we addressed the relevance in our discussion section, explaining how our findings can be applied to existing brief interventions.

Revised text:

“There are several existing *approaches and specific interventions* to reduce alcohol use, such as *Screening, Brief Intervention, and Referral to Treatment (SBIRT)*, cognitive behavioral therapy (CBT) or motivational enhancement therapy, that are considered to be appropriate for older adults” (See page 3)

“*First, practitioners should have attention for the elements that are related to the role of the practitioner in in-person interventions, more specifically in existing brief interventions which are already often used by practitioners.*” (See page 18)

Comment 6: Under 'Interpretation of findings', the authors state there are two important mechanisms that contribute to positive outcomes. The first is identified as 'awareness' but is not clear what the second is (perhaps knowledge?).

Reply: Thanks for this comment. Indeed, we didn't include the word mechanism. Moreover, we were not consistent in how we mentioned the second mechanism throughout the manuscript and therefore did some extra minor changes throughout the manuscript.

Revised text:

“Second, *accessible and* low-threshold participation is *an important mechanism* in interventions for older adults.” (See page 17)

We made textual changes when referring to the second mechanism accessible and low-threshold participation throughout the manuscript. (See page 2, 15, 17, 18)

Comment 7: Concerning the mechanism of awareness, some recent conceptual and empirical work has focused on this (termed 'problem recognition') in non-help seeking drinking groups [1, 2] which it may be worth highlighting. Notably, one recent study on problem recognition [3] also empirically supports the important point made about the threat of 'alcoholic' labeling undermining 'awareness'/problem recognition.

[1] James Morris, Ian P. Albery, Antony C. Moss, Nick Heather, Chapter 10 - Promoting problem recognition amongst harmful drinkers: A conceptual model for problem framing factors, Editor(s): Daniel Frings, Ian P. Albery, *The Handbook of Alcohol Use*, Academic Press, 2021
<https://doi.org/10.1016/B978-0-12-816720-5.00026-8>

[2] Jessica J. Smith, Panagiotis Spanakis, Rachael Gribble, Sharon A.M. Stevelink, Roberto J. Rona, Nicola T. Fear, Laura Goodwin, Prevalence of at-risk drinking recognition: A systematic review and meta-analysis, *Drug and Alcohol Dependence*, Volume 235, 2022,
<https://www.sciencedirect.com/science/article/pii/S0376871622001867>

[3] J. Morris, A.C. Moss, I.P. Albery, N. Heather, The "alcoholic other": Harmful drinkers resist problem recognition to manage identity threat, *Addictive Behaviors*, Volume 124, 2022
<https://doi.org/10.1016/j.addbeh.2021.107093>

Reply: We thank the reviewer for the suggestion to highlight these studies. We agree that referring to other publications about this subject can be informing and we recognize the importance of addressing literature about 'problem recognition' in people with problematic alcohol use.

Revised text:

"In addition, the current standard for responsible alcohol consumption is often known to only a small proportion among older adults (41), *and people with problematic alcohol use often do not recognize their own drinking pattern as problematic(42, 43).*" (See page 17)

Reviewer # 2

First off I wanted to state that this is important work in an underdeveloped area. I had a few concerns about the paper though:

Comment 8: Please make the research question more explicit so that readers can find it quickly and easily.

Reply: We want to thank the reviewer for sharing this suggestion. We agree that the text should be more specific about our research question. This also led us to the realization that the title, abstract and objective are not comprehensive. In order to present the focus of our study consistently, we adjusted the title and abstract.

Revised text:

"This study is set out to confirm, refuse, or refine the IPT, *consisting of the working elements, mechanisms and outcomes across six different contexts, as identified by Boumans et al. (22).* *More specifically, the objective of this study is to* understand from the perspective of professionals providing these interventions how (i.e., which elements), in which context, and why (which mechanisms) interventions are successful in reducing (problematic) alcohol use among older adults." (See page 4)

Because our study focuses on reducing problematic alcohol, we removed the word 'prevent' in the title and abstract. (See page 1)

Comment 9: Please describe the Initial Program Theory of Boumans et al in greater detail that you refer to. Readers will likely be unfamiliar with the work and it is problematic not describing it in greater detail on line 55.

Reply: We want to thank the reviewer for this suggestion, and changed some sentences. First, we mentioned that the authors of the review created a program theory based on their found working elements across different contexts. Second, we have mentioned more explicitly that the IPT used in this study is based on these findings (i.e. working elements and mechanisms) in these six contexts, as identified by Boumans et al. Third, we marked the working elements of the IPT separately with an asterisk in Table 3 to make it clearer for the reader which elements in our program theory originate from the IPT.

Revised text:

"This review explored what works for people of 18+ individuals years and older (which includes older adults), and older adults specifically, *and created an initial program theory* based on the found working elements across six different contexts." (See page 3)

“This study is set out to confirm, refuse, or refine *the IPT, consisting of the working elements, mechanisms and outcomes across six different contexts, as identified by Boumans et al (22).*” (See page 4)

Comment 10: Please fix the formatting in tables 2 and 3 so that they have greater readability. Table 2 had "professional" across three lines, this was the same for "alcohol use" and "relatives". This also goes for Table 3 such as "target group intervention".

Reply: We thank you for this pointing this out. We have revised the tables such that they are now easier to read. Table 2 is now on a separate page with landscape orientation and the cell sizes in Table 3 have been adjusted such that words are on two lines.

Revised text:

See Table 2 (page 8) and Table 3 (page 12-14).

Comment 11: In the results section, please label each italicized component as either a context, element, mechanism, or outcome. I believe that most of these are elements, but it would assist the reader in better understanding what each component is without confusing elements for mechanisms for example. This could be easily changed by writing for example on Line 51 "Professionals confirmed that the element (1) paying attention to drinking behavior..."

Reply: Thank you for this suggestion. It indeed makes the content easier to follow. We have not labelled each component with an (e), (m) or (o).

Revised text:

“Below, we describe the working elements (e), mechanisms (m) that contribute to reductions in alcohol use (o).” (See page 9)

We made changes throughout the result section to label each element (e), mechanism (m) and outcome (o). (See page 9-12)

Comment 12: Please define context, element, mechanism and outcome better on lines 44-49 for readers unfamiliar with this methodology. You put them together nicely but they still lack robust definitions.

Reply: We understand that providing more information about the Realist Evaluation can be helpful for readers who are not familiar with this approach. Therefore, we decided to provided more detailed information about this approach.

Revised text:

“Realist evaluation (RE) works from the assumption that interventions and their elements (E) work differently in different contexts (C). *The context refers to the circumstances in which interventions operate, e.g. an online context in which an intervention is provided.* Interventions may be successful in some contexts and not in others because the *underlying processes through which the interventions bring change, called the mechanisms (M),* are triggered to a different extent and lead to different *intervention outcomes (O).* *Therefore, the interaction between the intervention elements (E), contexts (C), and mechanisms (M) play an important role in shaping the intervention outcomes (O).*” (See page 4)

Comment 13: Table 3 is great, I just wanted to commend you all on how you put that information together.

Reply: We would like to thank the reviewer for this compliment.

VERSION 2 – REVIEW

REVIEWER	Morris, James London South Bank University
REVIEW RETURNED	23-Jan-2024
GENERAL COMMENTS	Congratulations on this paper and the amendments. A few very minor comments:

	In two incidents mechanisms are followed with (m) and in two incidents with (e).. Is this a mistake? In several places the manuscript refers to 'responsible' alcohol use. Ideally I suggest replacing this with 'lower risk' as there is arguably a greater degree of subjectivity and possibility of judgement (i.e., drinking above guidelines is 'irresponsible') associated with 'responsible'. As a possibly pedantic suggestion, it may be worth adding the letter identifiers (e,m,o) to the table headings for extra clarity. In the discussion, I suggest replacing the instances of the word 'minimally' with 'at least' for clarity. E.g., "were found in minimal in three contexts" would be clearer as "were found in at least three contexts"
--	--

REVIEWER	Fyffe, Ian Fraser Health Authority, Long Term Care and Assisted Living Research Unit
REVIEW RETURNED	01-Feb-2024

GENERAL COMMENTS	This has already been reviewed, so there is little for me to comment upon. However, I wanted to state that I appreciate this research given the subject matter and I believe it to be important. > On page 4 there was a typo of M for mechanism. An E was in place of the M. > On page 16 the word "providing" should be italicized next to "4)". > I was interested, though, as to what the average interview length was and the range of interview times. Especially because it was mentioned that interviewees may have wanted to leave early by agreeing with the IPT. Please mention this somewhere in the paper. Other than those minor revisions it was a great piece on an important and often overlooked topic.
---

VERSION 2 – AUTHOR RESPONSE

Comment 1: Please complete a thorough proofread of the text and correct any spelling and grammar errors that you identify.

Reply: We completed a proofread of the text, conducted by native English speakers.

Comment 2: Page 18, last paragraph - please italicise 'awareness' for consistency when referring to mechanisms, i.e. First, *awareness* is an important mechanism in preventing or reducing problematic alcohol use.

Reply: We thank you for pointing this out and we changed this in the manuscript.

Revised texts: First, *awareness* is an important mechanism in preventing or reducing problematic alcohol use. (See page 17)

Comment 3: Please check the citation on page 4, last paragraph, line 63 for Boumans et al. (22). In the reference list, Boumans et al. is number 21 and Ettner et al. is number 22.

Reply: Thank you for this comment. Indeed, we didn't use the correct citation. We have revised the text such that the citations are correct now.

Comment 4: In the Data collection section on page 7, please include the citation for Boumans et al (line 126).

Reply: Thank you for attending us on this. We included the citation.

Revised texts: At the end of the interview guide, we presented the previous found CEMO configurations of the IPT of the review of Boumans et al. (22), and invited participants to confirm, refute or refine the CEMOs and asked if they could comment on the CEMOs, see interview guide (see Supplementary Material 2).

Comment 5: To improve the structure of the introduction and to provide the reader with information on the Realist Evaluation and associated terminology at the earliest point before unfamiliar terms are mentioned, we recommend moving the paragraph starting 'Realist evaluations (RE) work from the assumption... Finally, based on the new data, the program theory may be defined' (currently on page 4, line 46 - 59) before the paragraph starting 'A systematic review which included 61 studies - using Realist Evaluation - focussed on...' (currently on page 3, line 19).

Reply: Thank you for this suggestion to improve the structure of the interview. We agree that moving this paragraph would improve readability, so we moved the paragraph as suggested.

Revised texts: (See page 2 and 3)

Comment 6: Thank you for updating the manuscript to describe the sampling method used. Since the COREQ checklist is included as a cited supplemental file for publication, please update the checklist 'Sampling' row so that it specifies convenience sampling and indicates the page where this is reported.

Reply: We thank you for pointing this out and we changed this in the COREQ checklist.

Revised texts: p.6, (See Supplementary material 1)

Response to reviewers

Reviewer #1

Congratulations on this paper and the amendments. A few very minor comments:

Reply: We would like to thank the reviewer for the detailed feedback. The feedback has improved our manuscript significantly!

Comment 7: In two incidents mechanisms are followed with (m) and in two incidents with (e).. Is this a mistake?

Reply: We thank the reviewer for pointing this out. It is indeed a mistake and we changed this in the manuscript. This also led us to the realization that we used lowercase letters (e) and (m) instead of capital letter (E) and (M) in the result section. To make sure the usage is consistent in the entire manuscript, we adjusted this in the result section.

Revised texts: Therefore, the interaction between the intervention elements (E), contexts (C), and mechanisms (M) play an important role in shaping the intervention outcomes (O). (See page 4)

Comment8: In several places the manuscript refers to 'responsible' alcohol use. Ideally I suggest replacing this with 'lower risk' as there is arguably a greater degree of subjectivity and possibility of judgement (i.e., drinking above guidelines is 'irresponsible') associated with 'responsible'.

Reply: Thank you for this suggestion. We agree that 'lower risk' suits better and therefore we replaced 'responsible' with 'lower risk'.

Revised texts: In addition, the current standard for lower risk alcohol consumption is often known to only a small proportion among older adults. (see page 17)

Comment 9: As a possibly pedantic suggestion, it may be worth adding the letter identifiers (e,m,o) to the table headings for extra clarity.

Reply: Thank you for this suggestion. We added the letter identifiers (c, e m, o) to Table 3 (See page 12) and Table 4 (see page 14)

Comment 10: In the discussion, I suggest replacing the instances of the word 'minimally' with 'at least' for clarity. E.g., "were found in minimal in three contexts" would be clearer as "were found in at least three contexts"

Reply: Thank you for this comment. We agree that changing this would be better.

Revised texts: Overall, five elements that were mentioned by more than one respondent were found in **at least three** contexts and can be labelled as general working elements (See page 15)

Moreover, two mechanisms that were mentioned by more than one respondent were found in **at least three** contexts (See page 15)

Reviewer #2

This has already been reviewed, so there is little for me to comment upon. However, I wanted to state that I appreciate this research given the subject matter and I believe it to be important.

Comment 11: On page 4 there was a typo of M for mechanism. An E was in place of the M.

Reply: We thank the reviewer for pointing this out. It is indeed a mistake and therefore we changed this in the manuscript.

Revised texts: Therefore, the interaction between the intervention elements (E), contexts (C), and mechanisms (M) play an important role in shaping the intervention outcomes (O). (See page 4)

Comment 12: On page 16 the word "providing" should be italicized next to "4)".

Reply: We thank the reviewer for pointing this out.

Revised texts: Overall, five elements that were mentioned (...) 4) **providing** *personalized content*

Comment 13: I was interested, though, as to what the average interview length was and the range of interview times. Especially because it was mentioned that interviewees may have wanted to leave early by agreeing with the IPT. Please mention this somewhere in the paper.

Reply: We thank the reviewer for this comment. The range of interview times is 40.4-85.4 minutes and the average is 59.3 minutes. We added this information on page 7.

Revised texts: Based on the participants preferences, online (80%) and telephone (20%) interviews were conducted between June and December 2022, **each lasting an average of 59.3 minutes** (ranging from 40.4 to 85.4 minutes).

Comment 14: Other than those minor revisions it was a great piece on an important and often overlooked topic.

Reply: We would like to thank the reviewer for this compliment and for the feedback, also in the previous round of feedback. It has improved our manuscript significantly!